# Intelligent Head-Mounted Obstacle Avoidance Wearable for the Blind and Visually Impaired [note 1]

**DOI:** 10.3390/s23239598

**Published:** 2023-12-04

**Authors:** Peijie Xu, Andy Song, Ke Wang

**Affiliations:** School of Computing Technologies, Royal Melbourne Institute of Technology (RMIT) University, Melbourne, VIC 3000, Australia; andy.song@rmit.edu.au (A.S.); ke.wang@rmit.edu.au (K.W.)

**Keywords:** intelligent wearable, obstacle avoidance, vision impaired, BVI, supervised machine learning

## Abstract

Individuals who are Blind and Visually Impaired (BVI) take significant risks and dangers on obstacles, particularly when they are unaccompanied. We propose an intelligent head-mount device to assist BVI people with this challenge. The objective of this study is to develop a computationally efficient mechanism that can effectively detect obstacles in real time and provide warnings. The learned model aims to be both reliable and compact so that it can be integrated into a wearable device with a small size. Additionally, it should be capable of handling natural head turns, which can generally impact the accuracy of readings from the device’s sensors. Over thirty models with different hyper-parameters were explored and their key metrics were compared to identify the most suitable model that strikes a balance between accuracy and real-time performance. Our study demonstrates the feasibility of a highly efficient wearable device that can assist BVI individuals in avoiding obstacles with a high level of accuracy.

## 1. Introduction

Approximately 43.3 million individuals were blind and 295 million people suffered from moderate to severe vision impairments, according to the estimates provided by the Global Vision Database 2019 Blindness and Vision Impairment Collaborators in 2020, and they are predicted to be 61 million and 474 million by 2050, respectively [1]. Vision loss not only limits mobility but also results in reduced engagement in daily activities, social participation, and the ability to detect hazards, leading to accidents, collisions, and falls [2,3]. One of the key difficulties they face is highly risky obstacles along their paths. The real-time detection of obstacles and hazards can greatly enhance their mobility and minimise their exposure to potential dangers. While traditional aids such as guide dogs and white canes are commonly used, they do not fully address the challenges of independent navigation [4]. Guide dogs are limited in their widespread adoption due to high costs and white canes are unable to detect obstacles beyond their range [5]. Many researchers and engineers have dedicated substantial effort to address this challenge.

In this study, we propose a smart and efficient wearable solution that can serve as a viable option for Blind and Visually Impaired (BVI) individuals. The human body’s centre of mass moves with each step, resulting in unavoidable head movements [6]; nevertheless, this intrinsic rotation of the head often introduces disruptions in sensor measurements. An intelligent device is introduced, which solves these natural head turns. The study focuses more on the performance of different algorithms when implemented on the hardware, rather than on algorithms at the software level only. In other words, it implies an emphasis on the integration of software and hardware. While the performance of different classifier algorithms is well known on general-purpose computers or GPUs, their performance on portable devices with limited computational resources involves tradeoffs. In obstacle avoidance scenarios, accuracy and real-time processing are the most critical factors, whilst at the same time, the critical hardware limitation, such as hardware resources available, needs to be satisfied as well. Various methods are evaluated and compared on a hardware device with constrained computational capabilities. The efficacy of the proposed method is proved by accurately identifying potential hazards while avoiding false alarms for harmless objects directly ahead. The detection system is fast and lightweight, operating directly on the device itself.

The paper is structured as follows: Section 2 reviews previous research on obstacle avoidance. In Section 3, a detailed description of the proposed head-mount smart device is presented. Section 4 and Section 5 outline the experimental setup, including the collection of sensor data via the wearable device in a controlled environment, the data labelling process, and the formulation and evaluation of learning algorithms. The results of the experiments are presented in Section 6, followed by the discussions in Section 7. Section 8 concludes the study and presents a vision for further research.

## 2. Literature Review

Numerous approaches have been proposed to provide environmental information to facilitate BVI individuals in navigating their daily activities in the past [7]. These approaches can be classified into Electronic Travel Aids (ETA), Electronic Orientation Aids (EOA), or Position Locator Devices (PLD). PLD is designed to provide position information for the individual using the device within the scene, while EOA assists BVI users in maintaining an accurate orientation during travelling. In this study, our main focus is to review the relevant literature in the field of ETA, which aims to perceive and interpret environmental information around the user. It aligns with the objective of our research.

The exploration of supplementing or replacing the white cane for BVI people can be traced back to the 1940s [8]. Ifukube et al. imitated the echolocation system of bats to detect small obstacles positioned in front of the user as early as 1991 [9]. The size constraint poses difficulties for users who wear the device for extended periods, and the ultrasonic sensor hardware used was also less reliable and durable at the time. Ran et al. introduced a wearable assistance system called “Drishti”, which facilitated dynamic interactions and adaptability to different environments [10]. However, the precision provided by the original equipment manufacturer’s ultrasound sensor was only 22 cm [10], which hardly meets the requirements for reliable real-time obstacle avoidance. “NavBelt” utilised eight ultrasonic sensors arranged on the abdomen to scan obstacles within a range of 120° [11]. The signals captured were subsequently processed via robotic obstacle avoidance algorithms [11]. A limitation of this system is its inability to reliably detect obstacles located above the user’s head. A navigation system that incorporated a memory function with an integral accelerometer was proposed to compute and record walking distance as a form of guidance [12]. The system included two vibrators and two ultrasonic sensors mounted on the user’s shoulders for obstacle detection, along with an additional ultrasonic sensor integrated into the white cane [12]. To complete the initial navigation route, it required the BVI user to carry a cane with a sighted individual accompanying them. This approach encountered challenges in effectively managing cumulative tracking errors, often causing a failure after a certain period of operation. The performance of ultrasonic sensor hardware has been increasing over the past few decades, which possesses high sensitivity and penetration capabilities, making them suitable for sensor-based solutions. Gao and colleagues developed a wearable virtual cane network with four ultrasonic sensors placed on the wrists, waist, and ankle of the user, respectively [13]. It is reported that the system can detect obstacles as small as 1 cm ${}^{2}$ located 0.7 m away [13]. Bouteraa provided a glass-framed assistive device supported by three ultrasonic sensors and a hand band equipped with a LiDAR sensor in a recent study [14]. It shows that the average walking time and the number of collisions are reduced compared to the conventional white cane used in their experiments, due to the integrated fuzzy decision support system [14]. However, the participants did collide with some small harmless obstacles during the validation experiment [14]. Both of the aforementioned studies incorporated the device with the upper limb, but the natural arm swing during walking was not considered.

A number of state-of-the-art techniques have gradually been integrated into developing wayfinding and mobile navigation assistance for BVI people with advancements in technologies such as edge computing, smart sensors, and AI. Related researchers and engineers attempted AR technology [15], RFID [16], and cloud systems [17,18,19] to develop an intelligent and commercially viable navigation mechanism or system. Nevertheless, these techniques often require significant computational or external support, such as 5G connectivity or e-tags, which can pose challenges in terms of resource requirements and dependence on external infrastructure. These challenges are reflected in network availability, latency, data loss or corruption, scalability, and congestion. For example, network outages or signal interference can disrupt the smooth functioning of navigation that depends on continuous communication. Timely data exchange is crucial in obstacle detection applications, where even a small delay in network communication can lead to serious consequences.

With the rapid proliferation of deep learning-based computer vision models, vision sensors have become increasingly popular in various systems. Those equipped with depth detection capabilities, RGB-D cameras, are the most popular. They relied on structured light or time of flight to circumvent the weaknesses of purely visual techniques by measuring the direct distance physically. Hicks et al. built a ski goggle with a depth camera, a small digital gyroscope, and an LED display to assist poor vision individuals for navigation by utilising their functional residual vision [20]. Aladrén et al. segmented obstacle-free pathways within the scenario by utilising depth and colour information captured via a consumer-grade RGB-D camera [21]. The unobstructed path segmentation algorithm operated at a rate of two Frames Per Second (FPS), while the overall system, involving range data processing, RGB image processing, and user interface generation, ran at a slower speed of 0.3 FPS on a laptop [21]. The method proposed by Yang et al. overcame the limitations of narrow depth field angles and sparse depth maps to improve close obstacle detection [22]. Their approach expanded the initial traversable area via a seeded growing region algorithm [22]. Lee and Medioni introduced a novel wearable navigation system combined with an RGB-D camera, a laptop, a smartphone user interface, and a haptic feedback vest [23]. The system estimated real-time ego motion and created a 2D probabilistic occupancy grid map to facilitate dynamic path planning and obstacle avoidance [23]. Diaz Toro et al. built a vision-based wearable system with a stereo camera for floor segmentation, occupancy grid built, obstacle avoidance, object detection, and path planning [24]. The system operated at 11 FPS, effectively handling both the floor segmentation and the construction of a 2D occupancy grid [24]. Takefuji et al. developed a system where a stereo camera is attached to clothing or a backpack [25]. The system employed the YOLO algorithm, enhanced with distance information, to calculate the direction of obstacles at 14 to 17 FPS [25]. The limitation of the system includes high energy consumption and a short battery life [25]. Xia et al. integrated a laser ranging chip with an image processing unit to enable the recognition of traffic lights, obstacle avoidance, and payment functionalities [26]. This system utilises a YOLO-lite algorithm; however, it is still significantly power-consuming with an average current of 226.92 mA per second [26]. A common challenge in all vision-sensor-based works mentioned above is the significant computational cost associated with carrying out the algorithm or device, which typically requires the user to have a high-performance laptop with them. However, a high-end laptop is potentially unaffordable for the BVI people as 90% of them live in low- and middle-income countries [27]. It is also impractical to walk around with a laptop all the time. Li et al. addressed the issue of high computational demands by cloud systems, but it necessitates confronting the substantial risks associated with the network as previously mentioned [19]. Additionally, the performance of cameras is significantly impacted under conditions of low illumination, such as during nighttime [26] or in low-light environments. This is due to their inherent sensitivity to lighting conditions, which are crucial for capturing clear images. We focus on investigating a feasible approach to create a lightweight and real-time solution. We aim to explore alternatives to traditional camera-based systems and computationally intensive computer vision algorithms and viable strategies that could potentially enable us to achieve real-time detection without a heavy computational burden.

## 3. Experimental System Setting

A head-mount smart device consisting of two interconnected modules, an acquisition module and a processing module, is proposed in this study, as illustrated in Figure 1 and Figure 2. The former perceives the surroundings by utilising ultrasonic sensors and a 9-DOF orientation Inertial Measurement Unit (IMU). The latter handles sensor control, data computation, and decision-making processes. A Raspberry Pi 4B is selected for the processing module due to its high portability and versatility. Unlike using a standard computer (laptop), this structure offers a compact and unobtrusive form factor, enhancing user comfort and freedom of movement. The ultrasonic sensors used in the prototype, the HC-SR04 and the MaxSonar-EZ1, weigh 8.7 g [28] and 4.23 g [29], respectively. As a result, the total sensor weight of our prototype does not exceed 90 g. Portable devices like the Raspberry Pi are engineered for optimised power consumption, enabling longer battery life and reduced reliance on external power sources. Nonetheless, employing such a low-profile, portable hardware platform does introduce certain challenges, such as the limited computational resources available compared to a computer. The Raspberry Pi features a 1.5 GHz quad-core ARM Cortex-A72 CPU, 4GB of LPDDR4 RAM, and multiple USB ports [30]. This constraint may impact the complexity and speed of data processing and decision-making algorithms. Balancing the need for timely detection with the hardware’s inherent limitations poses a significant challenge.

The ultrasonic sensors are arrayed in three rows, with a total of nine HC-SR04 (green in Figure 1) [28] in the first and third rows and a separate MaxSonar-EZ1 (yellow in Figure 1) [29] placed in the second row. Four sensors in the top row detect upper obstacles and the remaining five sensors in the bottom row are for lower regions. The advantages of these nine sensors are low cost and affordability with an effective detection angle of 15°, a maximum detection distance of 400 cm, and a minimum distance of 2 cm. They integrate both ultrasonic transmitters and receivers and exhibit an accuracy of up to 3 mm. The additional sensor in the middle row is to supplement the compensation and robustness of the sensor array, but it is a bit more expensive compared to the other sensors. It is featured with compensation for target size variations, well-balanced sensitivity, built-in noise reduction, real-time background auto-calibration, real-time waveform analysis, and noise rejection. It has a maximum range of 500 cm and communicates directly with the processing module via a USB port. Each sensor possesses a unique detection angle. Through such an ultrasonic sensor array, the collective detection range is similar to the binocular field of view. It allows for obstacle detection both in front and at the sides.

The user’s movement, including acceleration and rotation, is measured via the IMU, an Adafruit BNO085 [31] which integrates an accelerometer, gyroscope, and magnetometer. It transmits data from three axes of linear acceleration and the angular velocity of rotation around three spatial axes, as well as converts a four-point quaternion to the processing module via the I2C bus. The conversion of the quaternion or rotation vector is completed via IMU itself, which reduces the computational burden on the processing module and minimises the impact of drift errors.

## 4. Data

In this section, we provide a detailed account of how the data was collected, processed, and labelled for our obstacle avoidance system. The objective is to create an emulation that closely resembles real-life navigation scenarios while also having the capability to precisely control various factors for detailed investigations. The comprehensive data collection allowed us to effectively train and evaluate our models.

### 4.1. Data Collection

The data of the study is collected in a temperature- and humidity-controlled indoor environment, specifically within a 15 m long corridor. One side of the corridor features a clean wall, while the other side is furnished with long sofas. The floor is marked with parallel lines at distances of 1 m and 1.5 m from the walls. The spacing between these marks is either 0.5 m or 0.75 m, serving as guides for the participants, enabling them to walk at different paces and speeds. Additional angles of 75 and 60 degrees relative to the wall are marked to further enhance the realism of the study. It replicates situations where individuals deviate from the straight path or may need to navigate while directing their attention to the side. It is worth noting that this study focuses on the indoor environment. Whilst the outdoor environment introduces a significant amount of uncertainty and variations, it also has a higher fault tolerance for collisions. Indoor settings provide a more controlled environment with fewer variables and noises, which facilitates accurate validation of the prototype’s ability to recognise obstacles and address head movement issues. In addition, indoor scenarios are commonly more complex and crowded with a higher density of diverse obstacles per unit area. The complexity presents unique and significant challenges in obstacle detection, such as varied shapes, sizes, materials, and the effects of walls on sensor measurements, which are not as prevalent outdoors. Outdoor settings, while also varied, are generally more sparsely distributed.

Since there are no significant differences between individuals with full vision and BVIs in terms of walking speed and the deviation from a straight path [32], three healthy participants were invited to wear the head-mount device and introduced to adjust it to a comfortable position. The functionality of the device and the orientations of the sensors were verified to ensure the device was working properly. It is important to note that the positioning of the sensor set against the subject’s head is not guaranteed due to the variation in head sizes among different individuals. This variability is considered a form of noise, posing a challenge for subsequent model learning. The participants were then instructed to navigate through the designated route, which involved traversing the scenes from one side to the other and then returning. An example in Figure 3 is that the participant walks along the dotted line from side A to side B and then returns. Various variables were controlled and manipulated during the walk to create a comprehensive dataset. These variables included different head orientations (e.g., facing the front, facing the right, facing the left, and head turns), movement speeds (e.g., 0.5 m/s, 0.75 m/s, and 1 m/s), and the direction of walking (e.g., straight ahead or towards the wall at an angle). The BVI adults walked with a shorter stride length and a slower speed without obstacles compared to sighted people, at about 1.09 m/s [32]. However, Santos et al. reported that the BVI individuals walk faster than blindfolded when using the white cane in a scene with obstacles, at 0.4 m/s [3]. The deliberate selection of movement speeds, such as 0.5 m/s, 0.75 m/s, and 1 m/s, reflects a range of walking velocities that individuals typically exhibit in various contexts. These speeds cover a spectrum from leisurely walking to brisk walking, encompassing variations that the head-mount smart device may need to accommodate during real-time operation. The inherent and inevitable movements of the head are caused by the body’s centre of mass naturally shifting with steps [6]. The inclusion of distinct head orientations is designed to simulate this real-world scenario. Additionally, pedestrian encounter events were introduced, such as other pedestrian walks passing in the same direction or the opposite direction.

### 4.2. Data Processing

All sensory data were recorded with timestamps, but these data are not synchronised due to differences in the sampling frequencies of the sensors. We manually review the timestamped data, removing redundant and duplicate samples to ensure data validity. The noise data mentioned previously has not been artificially removed or cleaned as it ensures that the learning model remains effective even in the face of minor variations in the positioning of the sensors. Overall, a total of 10,234 valid data entries were collected from the three participants, forming the dataset for the subsequent stages of the study. Each entry comprises 20 distinct attributes, ten from the IMU and ten from the ultrasonic sensor array. The data extracted from the IMU includes three axes of linear acceleration, the angular velocity of rotation around three spatial axes, and four quaternion attributes. Figure 4 provides an enhanced visualisation of the data distribution on linear acceleration and rotational angular velocity. In this representation, the *x*-axis denotes the horizontal orientation of the cranial frame, reflecting vector acceleration and rotational angular velocity on both the left and right sides of the head. The *y*-axis symbolises the gravitational acceleration experienced by the user and the up and down rotation of the head, and the *z*-axis corresponds to the anterior–posterior axis. The outcomes of the data distribution are as expected. Specifically, the linear acceleration, with the exception of the horizontal axis (*x*-axis), exhibits a skewed distribution. This skewness signifies a positive acceleration attributable to vertical perturbations and forward propulsion, respectively, during the user’s ambulation. In contrast, the rotational angular velocities adhere to a normal distribution, signifying a stable rate of cranial rotation. The data from the ultrasonic sensor array are the distance readings of each ultrasonic sensor described in the previous section. On average, each second of movement generates more than 30 data entries.

### 4.3. Data Labelling

All data entries are labelled with negative and positive, indicating the absence or presence of obstacles that could pose a risk to a BVI person to ensure obtaining effectively trained models. In our case, an obstacle within a distance of 1.5 m from the person’s pathway is considered a positive case, representing a potential risk. Scenarios where obstacles are located more than 1.5 m away or are not directly in the person’s trajectory, such as those on the side or above, are labelled as negative. The distance of the label is based on the established literature in similar studies for BVI individuals [33,34].

## 5. Methodology

A real-time system is designed to respond to external input stimuli within a defined and limited period. Delay poses as the primary adversary in this context. Three major delays are challenging the system in our case, which are the processing delay in the sensory input system, the delay in presenting warning signals or recommendations to the BVI user, and the delay in the user’s response to these signals. This study focuses primarily on addressing the first two delays, as the third lies outside the scope of sensory computing. In other words, it is crucial to accomplish all tasks, including data acquisition, pre-processing, and decision making, within very short timeframes to ensure real-time performance. Hence, a parallel multithread processing approach is implemented in our system architecture.

The sensors operate at different frequencies, including sampling frequency, response frequency, and feedback frequency. It would be problematic if a particular was blocked by a task scheduled before it. Hence, various categories of sensor control are managed separately as a thread. The individual thread allows the sensors to operate in a synchronised manner, but without interfering with each other. Multithreading with a parallel mechanism effectively avoids the occurrences of task blocks and facilitates resource sharing during the process [35]. The device, such as the Raspberry Pi that we employ in this study, has limited computational resources like memory and CPU frequency. The threads share these scarce resources, which accelerate the system’s efficiency.

### 5.1. Model Learning

The trained model applied in the system needs to simultaneously satisfy the real-time capability for fast inference and the accuracy for an accurate detection of obstacles. The learning algorithm also needs to be able to handle sensor noises and unguaranteed variations in sensor placement due to different head sizes. We investigate the performance of various machine learning approaches, particularly a selection of widely recognised classification algorithms including (1) C4.5 Decision tree, (2) Naive Bayes, (3) k-Nearest Neighbour algorithm (kNN), and (4) Multilayer Perceptron (MLP). Additionally, we explore a range of ensemble methods that combine multiple classifiers, comprising (1) Bagging, (2) Random Forest, (3) AdaBoost, (4) Vote, and (5) Stacking. These ensemble algorithms generally enhance performance compared to using individual classifiers alone.

All the trained models are evaluated via ten-fold cross validation. It is a systematic technique that involves repeated holdout to pragmatically reduce the variance estimate [36]. Furthermore, we consider the impact of hyper-parameters, as the performance of most learning methods is associated with specific hyper-parameters (e.g., the size of leaf nodes in the Decision tree, the k value in kNN, and the layer connections in MLP). Prior empirical studies on each model are conducted to optimise the performance of these algorithms, aiming to identify the most effective combination of hyper-parameters. We set the batch size for all algorithms as 100 to ensure high-quality learning based on our empirical pre-study. The algorithms we used are from the Weka package, which supports all aforementioned learning methods and the ten-fold cross-validation evaluation.

To comprehensively compare the performance, our study incorporates both shallow and deep learning models as well. Deep learning has demonstrated exceptional capabilities in various complex machine learning tasks such as computer vision and natural language processing. We hence try to investigate how the deep learning models perform for a fast obstacle detection task. Several binary Neural Network (NN) classifiers are established for this purpose, including a shallow network (20-10-1) and three deep learning models. The former has one single hidden layer with 10 nodes, while the latter has two, ten, and fifty such hidden layers, respectively. The binary cross-entropy loss is utilised as the loss function in the networks, which is strongly coupled with the Sigmoid function. Stochastic gradient descent optimisation with a fixed learning rate is employed to control the amount of weight and bias adjustment during training. To evaluate the performance of these network models on our dataset, we also utilised ten-fold cross validation. The implementation of these deep learning models is based on PyTorch, one of the most widely used deep learning frameworks. PyTorch provides us with the capability to train our deep learning models efficiently using both GPUs and CPUs. It leverages an optimised tensor library, allowing us to take full advantage of the computational power offered by these hardware accelerators.

For our learning tasks, we employed a desktop computer equipped with an Intel Core i7-12700K processor, operating at a base frequency of 3.60 GHz. The computer has 16.0 GB of RAM and is equipped with an NVIDIA GTX 1060 GPU with 6 GB of dedicated memory. These hardware specifications provide sufficient computational power to perform the required learning and training tasks for our models. The WEKA version is 3.8.6 and the PyTorch version is 1.13.

### 5.2. Model Evaluation and Analysis

The performance of the aforementioned models is evaluated using several important metrics, including:Accuracy: measures the proportion of correctly classified instances (True Positive (TP) and True Negative (TN)) among all the predictions made by the model. It is calculated as (TP+TN)/(TP+TN+FP+FN), where FP is False Positive and FN is False Negative.Precision: quantifies how close the true positive predictions are to the positive instances in the dataset, indicating how many of the instances classified as obstacles are actual dangers ahead. It measures the ratio of true positives to the sum of true positives and false positives, i.e., TP/(TP+FP).Recall: evaluates the accuracy of the model’s positive predictions compared to the actual positive instances in the dataset, implying the proportion of predictions that are correctly classified as obstacles out of all the actual dangerous instances. It calculates the ratio of true positives to the sum of true positives and false negatives, i.e., TP/(TP+FN).F1-Score: provides a balanced measure of both precision and recall, reflecting the overall performance of the model. It evaluates the harmonic mean of precision and recall, i.e., 2TP/(2TP+FP+FN).Matthews Correlation Coefficient (MCC): treats the true class and predicted class as binary variables and calculates their correlation coefficient using the formula:
(1)$$MCC=\frac{\left(TP\times TN-FP\times FN\right)}{\sqrt{\left(TP+FP)(TP+FN)(TN+FP)(TN+FN\right)}}$$
By considering all the TP, TN, FP, and FN instances, MCC provides a more comprehensive measure of the correlation between the predicted and true classes [37]. It takes into account the balance between all four values and provides a more accurate assessment of the model’s performance in binary classification tasks [37].Mean Absolute Error (MAE): computes the average magnitude of errors or misclassifications made by the model, calculated via the sum of absolute errors divided by the total number of instances, $\Sigma_{i=0}^{n_{instances}-1}\left|error_{i}\right|/n_{instances}$, where $error_{i}$ represents the deviation from the model’s predictions.

While general machine learning metrics provide valuable insights, the study also focuses on assessing the models in terms of their ability to meet the specific requirements of real-time applications, including considerations such as operation-related time factors and resource utilisation factors. By incorporating these real-time performance aspects into the evaluation, the study provides a more comprehensive understanding of the models’ effectiveness and suitability for real-time scenarios.

The indicators for efficiency and durability observed include model Construction or training time (C time), model Execution CPU time (E time), model Prediction time (P time), and model size. Model execution time measures the CPU time taken by the trained models to make predictions on a given dataset, containing learning framework loading, model reading, and model prediction. The trained models are applied to a dataset of 500 entries on the Raspberry Pi, and the average CPU time of ten executions is calculated to determine the model execution time, providing insights into real-time prediction efficiency. Model prediction time focuses solely on the time spent predicting the dataset, excluding other activities like model loading. Model size refers to the amount of storage space that trained models require, which are binary files generated by different training frameworks. It is critical to consider the model size as larger models often consume more memory access, which is problematic due to the limited resources available on wearable devices used for data computing. Specifically, the evaluation of the models is conducted on a 64-bit Raspberry Pi OS with Linux kernel 5.10.

## 6. Experimental Results

We determine the most suitable combinations of hyper-parameters for the chosen classification algorithms, the top-performing ensemble methods, and the shallow and deep learning techniques by evaluating their accuracy performance. The accuracy of obstacle detection is a crucial factor in gauging the system’s dependability and effectiveness. After acquiring the optimal configurations, we proceed to assess other essential metrics for these models, aiming to gain a comprehensive understanding of their overall performance.

### 6.1. Hyper-Parameter Tuning

Various combinations of hyper-parameter settings for the Decision tree are tuned as Table 1, specifically the minimum number of instances per leaf ranging from 2, 10, and 50, and the confidence factor ranging from 0.25, 0.5, and 0.75. The confidence factor controls the pruning of branches in the Decision tree. Using a smaller confidence factor results in more aggressive pruning to a simpler and potentially less overfitted model. A larger confidence factor leads to less aggressive pruning, potentially capturing more details in the training data. However, it may also result in a more complex and overfitted model. It is observed that the model accuracy is positively correlated with the model size. The accuracy achieves 98.68% with a minimum of two instances per leaf and a confidence factor of 0.5. It can be observed that the training time is almost doubled with an increase in the confidence factor to 0.75, particularly when the minimum number of instances per leaf is as low as two. The optimal setting for the Decision tree is selected as 2 and 0.5 according to the comparison.

Table 2 illustrates the different configurations for the Naive Bayes model, including the inclusion and exclusion of a kernel estimator and supervised discretisation parameters. The model reaches an accuracy of 91.19% with a training time of 0.15 s when solely employing supervised discretisation. Disabling supervised discretisation results in a notable reduction in training time; however, it negatively impacts the model performance. When the kernel estimator is enabled, the model size increases by over 18 times, from 38 KB to 715 KB. The accuracy drops below 80% after disabling the kernel estimator.

The setting of kNN using the choice of distance measures (Euclidean, Manhattan, and Chebyshev) and different values of *k* (1, 3, and 5) is presented in Table 3. The value of *k* directly affects the bias–variance tradeoff in kNN. The choice of distance function determines how the similarity or dissimilarity between data points is calculated. The kNN algorithm does not require any training time, but the execution of the model involves all data instances. Consequently, the model size remains constant and undesirably large. The model achieves an impressive validation accuracy of 99.46% without significant overfitting, as proven via ten-fold cross validation. The lack of significant overfitting in ten-fold cross validation suggests that the model is not memorising the training data, but rather capturing meaningful patterns.

The results for the MLP are shown in Table 4. They are trained via standard backpropagation with the Sigmoid function. In the table, ‘a’ represents that the number of perceptrons is half the sum of the number of attributes and classes in the dataset in a hidden layer, which is ten perceptrons in each hidden layer in our study. Each increment of ‘a’ means an increment of the hidden layer. The training of the MLPs is obviously time-consuming. As more hidden layers are added to the network, the training time increases significantly, but this does not necessarily result in improved accuracy. Overall, the network with one hidden layer performs a 97.12% accuracy.

### 6.2. Analysis of Ensemble Algorithms

The study evaluates five ensemble algorithms. Bagging (Bootstrap aggregating) is a statistical estimation technique that helps reduce variance and mitigate overfitting. Random Forest consists of multiple Decision trees. AdaBoost (Adaptive Boosting) combines weighted sums of other learning algorithms to create a boosted classifier. Vote is a straightforward ensemble algorithm that trains multiple base estimators and makes predictions based on aggregating their outputs using weighted voting. Stacking is similar to Vote but with a different method of final aggregation. The base classifiers in Vote and Stacking in this study are four classifiers with the highest accuracy from the previous section (Bolded in each table of the previous model). The final aggregations of Vote and Stacking are averaged probabilities and the Decision tree model, respectively. Table 5 shows the result of the ensemble algorithms. Random Forest acquires the highest accuracy of 99.74% among them, but the size of this model increases significantly to 2486 KB.

### 6.3. Evaluation on Shallow and Deep Learning Methods

Deep learning networks with different numbers of hidden layers are trained and evaluated with ten-fold cross validation as shown in Table 6. The learning rate, batch size, and maximum epochs were optimised based on prior empirical studies. It is evident that the NN with one hidden layer outperforms the networks with more hidden layers, which is consistent with the findings in MLP. The deep learning network training incurs higher costs, despite the trained models being smaller compared to the MLP.

### 6.4. Comparative Analysis of Learning Methods

The best hyper-parameter tuned models that exhibit a balance between accuracy and speed are provided in Table 7, including accuracy, precision, recall, F1-Score, MCC, MAE, C time, E time, P time, and model size. The model that attained the highest performance in a specific metric is emphasised by being displayed in bold within its respective row.

## 7. Discussion

### 7.1. Model Comparison

The evaluation results of Table 7 are further summarised in Figure 5 and Figure 6. A total of 11 methods are included in the comparison. Figure 5 provides a comparison of different learning methods based on effectiveness and real-time metrics: accuracy, prediction time, and model size. The models are sorted based on the accuracy obtained from cross validation, and each model is represented by a different colour. Three subfigures in Figure 5 share the same *x*-axis. The height of each bar represents the average prediction time in the second subfigure as it is the mean prediction time of ten executions, and the standard deviation is indicated by the “I” symbol.

The leftmost six models in the first subfigure (Random Forest, Stacking, kNN, Vote, Bagging, and Decision tree) achieve an accuracy of more than 98%, but they differ considerably in terms of prediction time and model size. It takes no more than one second to predict the test dataset with 500 entries by three of the six methods, which are Random Forest, Bagging, and Decision tree. The remaining three methods all require more than 2.2 s to complete the test task. Achieving an inference time of 2.2 s does not imply that the models are incapable of real-time prediction, as this time refers to processing 500 entries. The number of entries coming in is typically over 30 entries per second, which is significantly lower than 500. Nevertheless, the models with faster prediction times are generally preferred in practical applications, as the real-time performance is still affected by the other two delays mentioned in the previous section.

When it comes to model size, Random Forest falls short in comparison. It has the largest size, exceeding 2 MB. The memory requirement of Random Forest may remain manageable for devices like the Raspberry Pi, which possesses reasonable capabilities. However, it poses significant challenges when attempting to port the detection system to a more cost-effective and compact processing unit, such as a microcontroller. Microcontrollers generally have limited computational power and memory capacity, but also extremely lower costs. The larger memory demands of certain models might become impractical or even infeasible to implement on such constrained hardware as prediction is just one of the components in obstacle detection. In contrast, the Bagging model is only 237 KB and the Decision tree is a mere 53 KB.

The MLP ranks seventh in accuracy at 97.12% with a smaller model size compared to the Bagging and the Decision tree. Its prediction time is longer than the other two models, even twice as long as that of the Decision tree. The remaining four models, Naive Bayes, two networks, and AdaBoost exhibit favourable characteristics such as small model sizes and fast prediction time, all completing in under 1 s. Notably, the exceptionally satisfactory model sizes achieved via both NNs can be attributed to the differences in their training frameworks. It is likely that the more efficient training frameworks they utilised led to shorter model loading times and optimised model storage methods. Efficient training frameworks can enhance the model’s compression capabilities, reducing its size without significantly compromising the performance. However, the accuracies of all four models fall within a relatively lower range, ranging from 83.39% to 91.19%. While their efficiency in terms of size and speed is commendable, the emphasis is placed on higher accuracy levels to ensure the system’s reliability and effectiveness in real-world scenarios. Consequently, other models that offer a better balance between accuracy and efficiency were selected instead of these models.

Figure 6 shows the comparison against the other performance metrics including precision, recall, F1-score, MCC, and MAE. The *y*-axis maintains the same order as depicted in Figure 5. Each metric is represented using the same colour scheme, ensuring consistency and easy comparison across the different models or methods. Among the first six models, all of them exhibit remarkably satisfactory precision, recall, and F1-score. In applications like obstacle detection, recall plays a vital role among the metrics as it represents the model’s ability to correctly categorise hazards by identifying the proportion of all actual hazards accurately. Obstacles that exist but go undetected are indicated via FN. Minimising FN is crucial in scenarios where safety is critical, as it ensures that the system effectively identifies and responds to potential hazards to minimise the risk of accidents or mishaps. Both Bagging and Decision tree models exhibit a slightly lower yet competitive performance compared to the top four models in terms of MCC. The performance metrics of the remaining five models have a considerable gap between the top six models.

Therefore, both Bagging and Decision tree models still perform admirably and are comprehensive among the top choices, despite not achieving the highest accuracy. The Decision tree stands out, not only due to its great effectiveness and suitability but also because it can be readily converted into a set of selection statements out of the training framework, leading to further reductions in running time and size. This capability makes it particularly appealing and advantageous in various scenarios. Of course, despite the lower performance of the remaining models, they can still serve as a valuable solution bank for various scenarios and applications. By maintaining a solution bank with multiple models, one can tailor the selection based on the specific context and needs of each situation. For example, the Random Forest model, which achieved the highest accuracy, can be an excellent choice when computational resources and power are abundant.

### 7.2. Explainability of the Model

To enhance our comprehension of the discriminatory capabilities of the Decision tree model in detecting the presence of obstacles, an additional small model was subjected to training. In this particular model, we have configured the minimum number of instances per leaf to be 50, and the confidence factor has been set at 0.5. Consequently, a Decision tree comprising 29 terminal nodes has been derived. The size of this tree is a mere 15 KB, yielding an accuracy rate of 94.44%. In contrast to the prior Decision tree model, this model exhibits a reduction in size exceeding two-thirds, while also experiencing only a marginal decline in accuracy, specifically by 4.24%. Figure 7 presents a graphical representation of this Decision tree model. All positive nodes are shown in black boxes. The initial decision node of this model is partitioned and eliminated based on the condition ‘btm5 <= 71.731’, where ‘btm5’ denotes the distance reading furthest to the right on the lower side of the ultrasonic array within the device. When the feature ‘btm5’ exhibits a value less than or equal to 71.731, the subsequent decision nodes evaluate the values of ‘quat_j’ and ‘quat_i’. Here, ‘quat_j’ and ‘quat_i’ represent the quaternion vectors associated with the user’s head rotation. These three conditional assessments collectively indicate that the device conducts an assessment for potential obstacles situated to the right of the user’s head, while also considering the head’s orientation. In cases where the available data did not provide sufficient evidence of rightward head rotation, the system subsequently initiated a judgment of ‘btm2’. The ‘btm2’ denotes the second distance reading obtained from the left side of the lower ultrasonic array. It is pertinent to highlight that the construction of this streamlined Decision tree did not incorporate all available features; specifically, only 13 out of the 20 attributes were employed. Among these, notable exclusions encompassed linear accelerations along both the vertical (*y*-axis) and forward (*z*-axis) directions, as well as rotational angular velocity along the horizontal (*x*-axis) and *z*-axis. The rationale behind this omission is rooted in the observation that these particular data components are integrated into higher-dimensional quaternion attributes. In other words, the head rotation speed and the barycenter during walking have a comparatively minor impact on obstacle detection. It can be seen that the Decision tree model excludes or determines the presence of obstacles in different directions by sequentially evaluating conditions and features in the tree structure until it reaches a leaf node that provides a prediction or classification regarding the presence of obstacles.

### 7.3. Real-Time Detection

To close the loop of the study, we developed an operational prototype capable of continuous obstacle detection on the sensor input as long as power is available. The system is designed to transform a positive detection outcome into an audio signal, providing real-time feedback to the user. When an obstacle is detected, the system emits a slightly prolonged beep sound as a warning signal. If the obstacle persists 0.5 s after the initial beep, another beep is triggered, ensuring ongoing alerts until no further positive detection. Tests conducted on volunteers with the proposed wearable setup demonstrated accurate and timely detection, even while individuals were moving and looking around.

## 8. Conclusions

This study aims to develop a practical, efficient, and accurate obstacle avoidance method for BVI individuals using an intelligent head-mount device. We propose a mechanism that combines an ultrasound sensor array and IMU sensor to detect obstacles directly in front of the person, excluding those on the sides, as head turns could affect sensor readings. The goal is to provide real-time obstacle detection on the person’s path, aiding them in navigation. The results demonstrate the effectiveness of the proposed method. It achieves high accuracy in detecting obstacles directly in front of the person, even when they are looking sideways or exhibit slight deviations in wearing the device from the standard position. Additionally, the model size is small enough to enable real-time performance on a resource-limited device. By choosing ultrasound and IMU sensors, we achieved a compact device, making the proposed method energy efficient and ideal for assisting BVI individuals. Among the various methods tested, the Decision tree emerges as the most suitable choice due to its optimal combination of accuracy and speed for the BVI assistant task. More sophisticated methods are proved suboptimal due to various relevant costs.

In our future study, we aim to enhance the proposed system to broaden its applicability and performance. One foreseeable improvement is to incorporate more complex indoor settings to assess the system’s adaptability and robustness, introducing scenarios with furniture scattered around and the presence of moving dynamic objects. Furthermore, we will explore the integration of additional types of sensors, such as low-cost vision sensors and temperature sensors, to compensate for variations in detection and provide a more comprehensive understanding of the surroundings. This multi-sensor approach will help enhance the system’s accuracy and reliability.

## Figures and Tables

**Figure 1 sensors-23-09598-f001:**
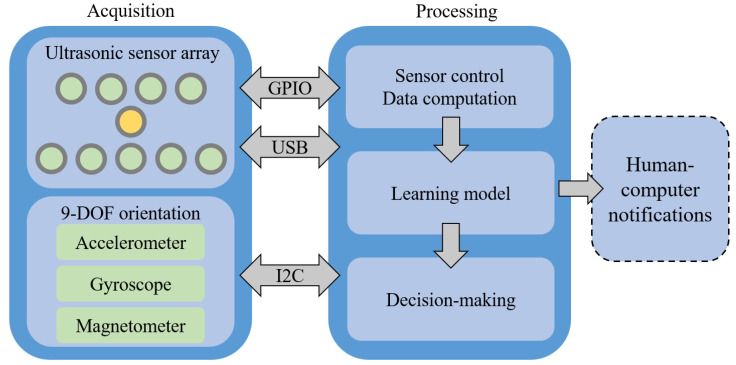
Experimental System Setting.

**Figure 2 sensors-23-09598-f002:**
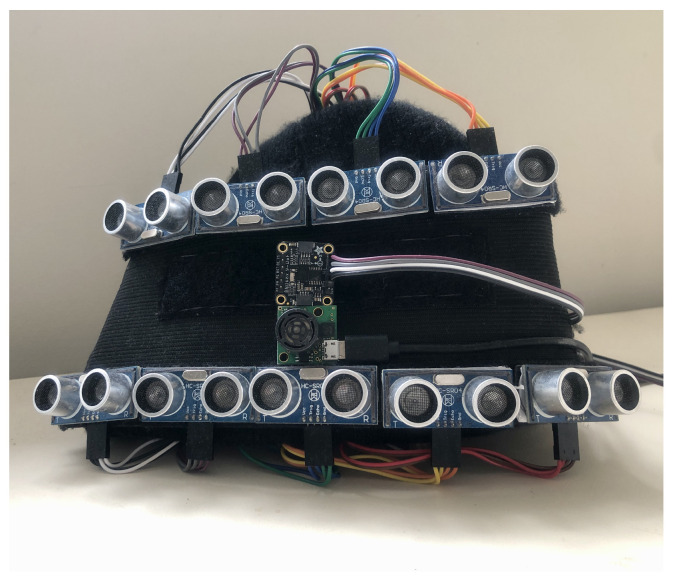
The photo of the device.

**Figure 3 sensors-23-09598-f003:**
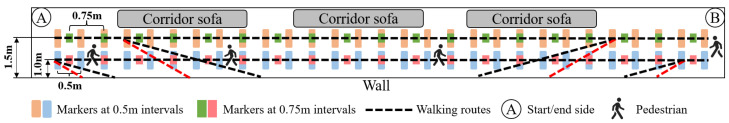
The scenario setting for data collection.

**Figure 4 sensors-23-09598-f004:**
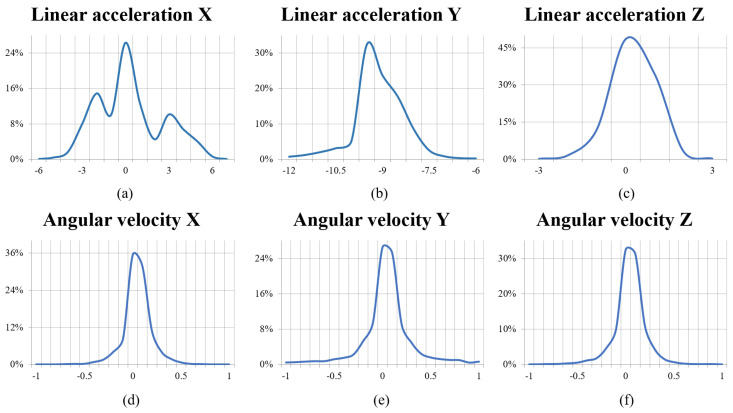
The visualisation of the sensory data distribution: (**a**) The vector acceleration in the horizontal direction; (**b**) The gravitational acceleration in the vertical direction; (**c**) The vector acceleration in the forward and backward direction; (**d**) The rotational angular velocity in the horizontal direction (Left and right rotation of the head); (**e**) The rotational angular velocity in the vertical direction (Up and down rotation of the head); (**f**) The rotational angular velocity in the forward and backward direction.

**Figure 5 sensors-23-09598-f005:**
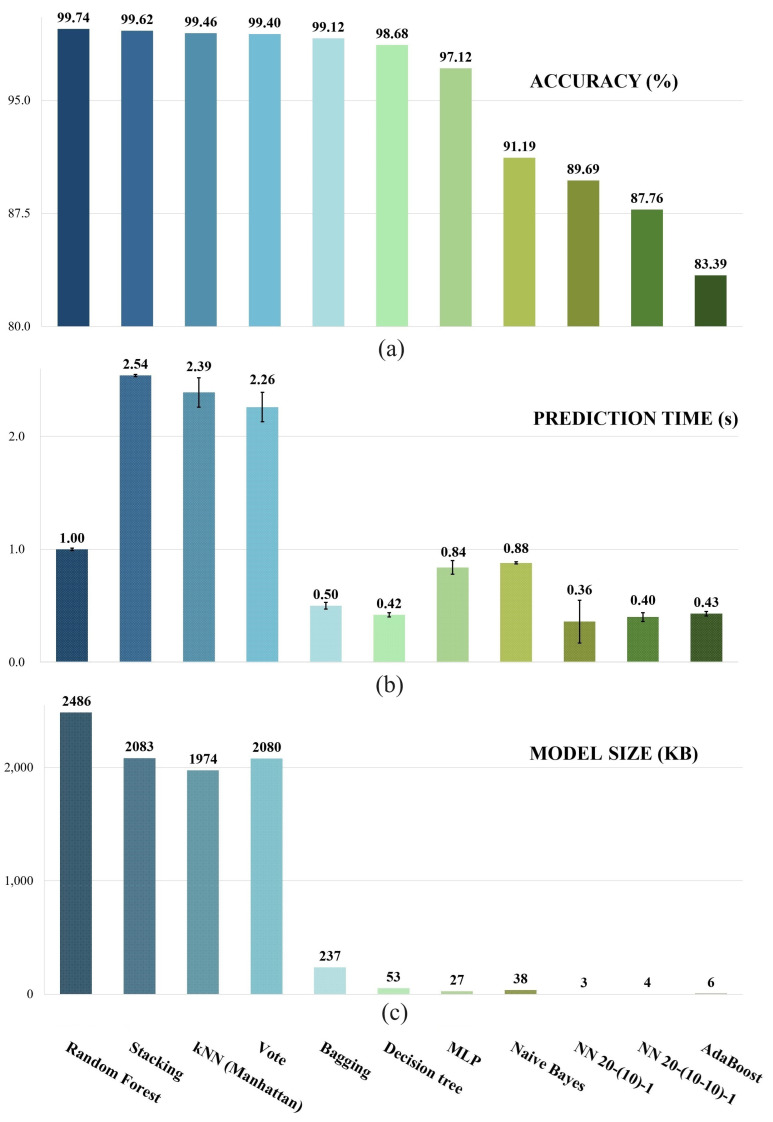
Effectiveness metrics: (**a**) Accuracy performance; (**b**) Prediction time; (**c**) Model sizes.

**Figure 6 sensors-23-09598-f006:**
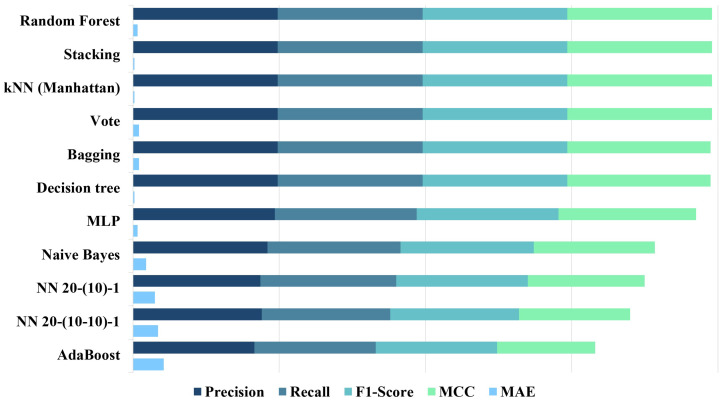
Additional performance metrics.

**Figure 7 sensors-23-09598-f007:**
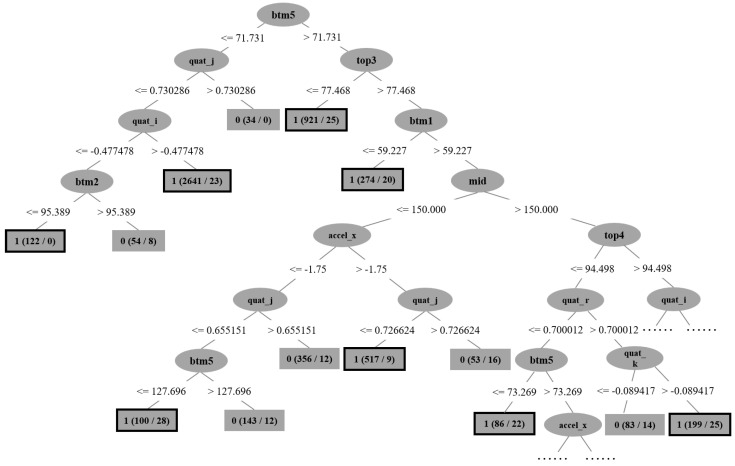
Visual representation of an example Decision tree.

**Table 1 sensors-23-09598-t001:** Example of hyper-parameter tuning—Decision Tree.

Min. Num. of Objects	Confidence Factor	Accuracy (%)	Construction Time (s)	Model Size (KB) (# of Leaves/Tree Size)
2	0.25	98.64	0.14	50 (127/253)
**2**	**0.5**	**98.68**	**0.14**	**53 (134/267)**
2	0.75	98.66	0.33	53 (134/267)
10	0.25	97.41	0.14	29 (68/135)
10	0.5	97.42	0.14	29 (69/137)
10	0.75	97.42	0.26	30 (71/141)
50	0.25	94.38	0.13	15 (29/57)
50	0.5	94.44	0.13	15 (29/57)
50	0.75	94.44	0.17	15 (29/57)

**Table 2 sensors-23-09598-t002:** Example of hyper-parameter tuning—Naive Bayes.

Use Kernel Estimator	Supervised Discretisation	Accuracy (%)	Construction Time (s)	Model Size (KB)
False	False	79.74	0.04	7
True	False	89.25	0.05	715
**False**	**True**	**91.19**	**0.15**	**38**

**Table 3 sensors-23-09598-t003:** Example of hyper-parameter tuning—kNN.

Distance Function	k Value	Accuracy (%)	Model Size (KB)
Euclidean	1	99.11	1974
Euclidean	3	97.40	1974
Euclidean	5	96.98	1974
**Manhattan**	**1**	**99.46**	**1974**
Manhattan	3	98.29	1974
Manhattan	5	98.14	1974
Chebyshev	1	98.04	1974
Chebyshev	3	94.16	1974
Chebyshev	5	93.30	1974

**Table 4 sensors-23-09598-t004:** Example of hyper-parameter tuning—MLP.

Hidden Layers Setting	Number of Hidden Layers	Accuracy (%)	Construction Time (s)	Model Size (KB)
**‘a’**	**1**	**97.12**	**13.05**	**27**
‘a, a’	2	96.67	22.01	36
‘a’ × 10	10	49.89	113.58	101
‘a’ × 50	50	49.89	641.67	427

**Table 5 sensors-23-09598-t005:** Example of ensemble algorithms.

Learning Algorithm	Accuracy (%)	Construction Time (s)	Model Size (KB)
Bagging	99.12	0.56	237
**Random Forest**	**99.74**	**1.89**	**2486**
AdaBoost	83.39	0.26	6
Vote	99.40	11.70	2080
Stacking	99.62	118.54	2083

**Table 6 sensors-23-09598-t006:** Evaluation on shallow and deep learning methods.

Networks	Number of Hidden Layers	Accuracy (%)	Construction Time (s)	Model Size (KB)
**20-(10)-1**	**1**	**89.69**	**1241.75**	**3**
20-(10-10)-1	2	87.76	1342.82	4
20-(10-…-10)-1	10	87.47	1814.44	11
20-(10-…-10)-1	50	84.46	2168.99	51

**Table 7 sensors-23-09598-t007:** Evaluation on learning methods.

Learned Models	Acc (%)	Pre	Rec	F1	MCC	MAE	C Time (s)	E Time (s)	P Time (s)	Size (KB)
Decision tree	98.68	**0.99**	**0.99**	**0.99**	0.98	**0.01**	**0.14**	14.48 ± 0.98	0.42 ± 0.02	53
Naive Bayes	91.19	0.92	0.91	0.91	0.83	0.09	0.15	15.36 ± 0.23	0.88 ± 0.04	38
kNN (Manhattan)	99.46	**0.99**	**0.99**	**0.99**	**0.99**	**0.01**	N/A	18.74 ± 1.35	2.39 ± 0.19	1974
MLP	97.12	0.97	0.97	0.97	0.94	0.03	13.05	15.43 ± 0.23	0.84 ± 0.01	27
Bagging	99.12	**0.99**	**0.99**	**0.99**	0.98	0.04	0.56	14.33 ± 0.67	0.50 ± 0.06	237
Random Forest	**99.74**	**0.99**	**0.99**	**0.99**	**0.99**	0.03	1.89	17.88 ± 0.59	1.00 ± 0.02	2486
AdaBoost	83.39	0.83	0.83	0.83	0.67	0.21	0.26	13.48 ± 0.53	0.43 ± 0.03	6
Vote	99.40	**0.99**	**0.99**	**0.99**	**0.99**	0.04	11.70	20.99 ± 1.09	2.26 ± 0.13	2080
Stacking	99.62	**0.99**	**0.99**	**0.99**	**0.99**	**0.01**	118.54	20.89 ± 1.27	2.54 ± 0.13	2083
NN 20-(10)-1	89.69	0.87	0.93	0.90	0.80	0.15	1241.75	**3.04 ± 0.02**	**0.36 ± 0.01**	**3**
NN 20-(10-10)-1	87.76	0.88	0.88	0.88	0.76	0.17	1342.82	3.11 ± 0.02	0.40 ± 0.01	4

**Abbreviations**: Acc, Accuracy; Pre, Precision; Rec, Recall; F1, F1-Score; MCC, Matthews Correlation Coefficient; MAE, Mean Absolute Error; C time, model Construction or training time; E time, model Execution CPU time; P time, model Prediction time.

## Data Availability

The data are not publicly available due to privacy.

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
