# Peer review of "Intelligent Head-Mounted Obstacle Avoidance Wearable for the Blind and Visually Impairedâ€"

_sensors, 2023, doi:10.3390/s23239598_

Round 1
Reviewer 1 Report
Comments and Suggestions for Authors
This article introduces an intelligent head-worn obstacle avoidance device designed to assist blind and visually impaired individuals in mitigating the risks associated with obstacles during independent mobility. By employing real-time obstacle detection and warning mechanisms, this device substantially enhances their ability to navigate and reduces their exposure to potential hazards. The device employs an intelligent approach to address the impact of natural head movements on sensor measurement accuracy and accurately identifies potential hazardous objects, thus minimizing false alarms for benign objects. The device is characterized by its swiftness, portability, and operates directly on the user's device. Nevertheless, there remain certain issues within this article that require further investigation and revision. The following comments delineate areas for improvement.
1. The test data is limited to indoor environments, lacking outdoor scenarios, and the testing environment is overly simplistic, thus limiting its applicability to real-life situations of the BVI population. It is advisable to configure the testing environment to more closely resemble the real-life situations encountered by BVI patients.
2. The article mentions that the wearable device is "lightweight" and "small in size," but from Figure 2, it appears that the device may not fully meet these characteristics. The device's hardware modules could be more integrated, compact, and lightweight to better align with users' daily wearability requirements.
3. All nine ultrasonic sensors used in the device are positioned on the front, implying that the system can only detect obstacles directly in front of the user and may not effectively avoid obstacles from other directions. Can this meet the requirements for "obstacle avoidance"? Is there room for improvement in hardware and software algorithms?
4. Images provided in the article, when combined in multiple-image compositions, should be clearly labeled individually for ease of reference.
5. The "The scenario setting for data collection" diagram in Figure 3 is only a schematic representation and does not provide actual images of the test environment. This results in a lack of reliability in the testing process.
6. The device's obstacle avoidance notifications are limited to "beep" sounds. Is this level of notification content too simplistic? Could additional information be included, such as the distance of obstacles from the user, the direction of obstacles, or the size of obstacles, to better assist users in avoiding obstacles?
7. The article emphasizes that the system is a "Real-time Detection" aimed at " transform a positive detection outcome into an audio signal, providing real-time feedback to the user." However, it does not further elucidate the relationship between the system's chosen model algorithm and the accuracy of obstacle avoidance in real-world scenarios, nor does it provide comparative data with other works to intuitively demonstrate the superiority of this work's algorithms.
8. The article mentions that the " the positioning of the sensor set against the subject’s head is not guaranteed." Could this affect the accuracy of experimental results? Has this issue been considered in the study?
9. The target population of this project is "blind and visually impaired (BVI) individuals," yet testing involved only "three healthy participants." Is there a disparity between these test participants and real-world BVI users? Why were blind and visually impaired individuals not invited to participate in the testing to obtain more realistic data?
Comments on the Quality of English LanguageNone
Author Response
Dear Editor/Reviewer,
Thank you for processing our manuscript and providing us with an opportunity to address the reviewers’ comments.
We are uploading (a) our point-by-point response to the comments (below) (response to reviewers), (b) an updated manuscript with changes highlighted in red (Supplementary Material for Review), and (c) a clean updated manuscript without highlights (Main Manuscript).
Best regards,
Peijie Xu et al.

Reviewer 2 Report
Comments and Suggestions for Authors
The authors propose an intelligent head-mount device to assist BVI people with this challenge. The objective of this study is to develop a computationally efficient mechanism that can effectively detect obstacles in real-time and provide warnings. The learned model aims to be both reliable and compact so that it can be integrated into the wearable device with small size. In this study, over thirty models with different hyper-parameters were explored by the authors and their key metrics were compared to identify the most suitable model that strikes a balance between accuracy and real-time performance. This study demonstrates the feasibility of a highly efficient wearable device that can assist BVI individuals in avoiding obstacles with a high level of accuracy.
The paper presents good work but below are my concerns:
1. Introduction and survey part is short. Add few more references and include a table.
2. Change section 3 title as System Model or anything suitable instead of System Built
3. I think data processing shall be added after describing the methodology. I mean, swap section 4 and 5 if required.
4. Couldn't infer anything from figure 7
5. I strongly feel, the work missed out the comparison of the novelty with existing system. Carrying out the comparison with RF, KNN, MLP etc may give diff results but I am finding something missing here strongly. Please check it out.
Comments on the Quality of English LanguageMinor editing of English language required
Author Response

(The authors gave the same response as above.)

Reviewer 3 Report
Comments and Suggestions for Authors
I don´t have special suggestions for authors, just some comments. The paper is well structured, including literature review, methodology , experiments, statistical evaluation, processing results and discussion. Authors present a well done research work devoted to an innovation of wearable assistive technology - a head mounted device for blind and visually impaired persons to assist while walking alone in an internal environment. Their main focus in literature review was to identify relevant literature in the field of ETA - Electronic Travel Aids.
The presented work offers a laboratory device using relatively cheap solutions - a mechanism that combines ultrasound sensor array and IMU sensors to detect obstacles directly in front of the person, but in combination with the advanced processing technology (AI, NN for accurate detection of obstacles, statistical models for analysis, ...) offer an efficient and reliable results. Experiments were provided only with 3 probands. It is a number that may be acceptable for starting experiments to prove the methodology and device design. They provided also analysis of 11 different learning methods based on effectiveness and suitability metrics: accuracy, prediction time and model size.
They plan to continue in their research by enhancing the proposed system to broaden its applicability and performance adding also presence of moving objects. It will require to integrate additional types of sensors.
Author Response

(The authors gave the same response as above.)

Round 2
Reviewer 1 Report
Comments and Suggestions for Authors
After the revison, the article can be accepted.
Comments on the Quality of English LanguageNone
Reviewer 2 Report
Comments and Suggestions for Authors
After carefully examining the response to my comments and the revised version, I am satisfied with the revisions. Hence, the paper shall be accepted in present form.